# Single-digit-micrometer thickness wood speaker

Wentao Gan[1,3], Chaoji Chen [1,3], Hyun-Tae Kim [2,3], Zhiwei Lin[1], Jiaqi Dai[1], Zhihua Dong[1], Zhan Zhou[1], Weiwei Ping[1], Shuaiming He[1], Shaoliang Xiao[1], Miao Yu [2]* & Liangbing Hu [1]*

Thin films of several microns in thickness are ubiquitously used in packaging, electronics, and acoustic sensors. Here we demonstrate that natural wood can be directly converted into an ultrathin film with a record-small thickness of less than 10 µm through partial delignification followed by densification. Benefiting from this aligned and laminated structure, the ultrathin wood film exhibits excellent mechanical properties with a high tensile strength of 342 MPa and a Young's modulus of 43.6 GPa, respectively. The material's ultrathin thickness and exceptional mechanical strength enable excellent acoustic properties with a 1.83-times higher resonance frequency and a 1.25-times greater displacement amplitude than a commercial polypropylene diaphragm found in an audio speaker. As a proof-of-concept, we directly use the ultrathin wood film as a diaphragm in a real speaker that can output music. The ultrathin wood film with excellent mechanical property and acoustic performance is a promising candidate for next-generation acoustic speakers.

[1] Department of Materials Science and Engineering, University of Maryland, College Park, MD 20742, USA. [2] Department of Mechanical Engineering, University of Maryland, College Park, MD 20742, USA. [3] These authors contributed equally: Wentao Gan, Chaoji Chen, Hyun-Tae Kim. *email: mmyu@umd.edu; binghu@umd.edu

Thin films of several microns or even nanometers in thickness are used in a wide range of applications, including solar cells[1], food packaging[2], water treatment[3], personal electronics[4,5], and acoustic sensors[6]. Acoustic membranes, for example, are generally very thin (micron-scale) and must be mechanically robust with a high modulus to enable a highly sensitive frequency response and high vibrational amplitude. In the past few decades, tremendous efforts have been dedicated to developing various acoustic thin film materials based on plastic[7], metal[8], ceramic[9], and carbon-based materials[10–12] for the purpose of enhancing the quality of the sound output. In particular, plastic thin films are used ubiquitously in commercial speakers as they are low cost and easy to process with controllable thickness and high modulus. However, most plastic films are difficult to degrade, creating an enormous impact on the environment[13–15]. Furthermore, metal, ceramic, and carbon-based materials demonstrate higher modulus than plastic films, which improves the frequency response of the acoustic membrane. But these compounds are typically higher cost and require complex manufacturing processes that consume large amounts of energy, limiting scalable applications. Therefore, it is desirable, yet highly challenging to prepare a high-performance and biodegradable acoustic thin film in a green and more cost-effective way.

With the emphasis on environmental protection in recent years, natural cellulose-based materials, such as bagasse[16], wood fibers[17–20], chitin[21,22], cotton[23–25], bacterial cellulose[26], and lignocellulose[27,28], have provided an environmentally friendly and rapid way to synthesize thin films from sustainable materials rather than using limited fossil resources. In addition to the advantages of abundance and renewability, cellulose fibers display extraordinary mechanical properties with a theoretical tensile strength up to 7.5 GPa and a Young's modulus up to 120 GPa, which are even higher than those of common metals, ceramics, and many composites[29,30]. Most cellulose-based materials are generally prepared from bottom-up approaches that involve first breaking down the cellulose fibers by mechanical[31], chemical[32], or biological[33] methods, followed by reconstruction into thin films via filtration, freeze-drying, stretching, or slurry casting. Although the advantages of these bottom-up methods include the ability to control the fiber length and the film structure, the multiple-step manufacturing process consumes large amounts of water, reagents, energy, and time, hindering practical applications. Additionally, it is difficult to maintain the excellent mechanical properties of the elementary cellulose fibers in bottom-up assembled cellulose-based films due to the resulting disordered arrangement of the fibers[34–38].

The natural structure of wood provides an effective alternative for more scalable and mechanically robust cellulose films. Featuring aligned cellulose fibers embedded in a soft hydrogel matrix of lignin and hemicellulose within a porous channel structure that runs along the longitudinal direction (Fig. 1a)[39–45], wood can be utilized as a scaffold for constructing cellulosic thin films via top-down approaches that are more scalable and cost-effective compared to bottom-up methods. The cellulose fibers biosynthesized in the wood cell walls dominate the mechanical properties of wood, while lignin and hemicellulose act as reinforcement agents that tightly bind the cellulose fibers together. Maintaining the alignment of the cellulose fibers rather than breaking them down enables robust mechanical properties of the resulting material. Furthermore, as the most abundant biomass on Earth, wood is renewable and biodegradable, making it more environmentally sustainable than plastics or metals. However, thin cellulosic films made from wood generally have a large thickness of more than 80 μm[27,46] far from the minimum thickness required for high-performance acoustic application.

Here, we report a facile and large-scale top-down approach for fabricating an ultrathin wood film with a thickness as low as 8.5 μm using a partial delignification and densification process (Fig. 1a). Our approach to synthesizing this membrane involves the partial removal of lignin and hemicellulose from natural balsa wood to generate a higher porosity material while retaining most of the cellulose in the cell walls, which then allows us to densify the treated wood by hot-pressing for a thickness reduction of ~97% (from 300 μm to 8.5 μm, Fig. 1b, Supplementary Fig. 1). Meanwhile, the highly aligned cellulose fibers are densified, which greatly enhances the hydrogen bond formation between the neighboring cellulose molecular chains. The densely packed wood cell wall structure and the highly aligned cellulose fibers contribute to a superior tensile strength and high Young's modulus. Furthermore, using industry-adopted cutting methods, we are able to fabricate a meter-long natural balsa wood film in the lab (Fig. 1c), revealing this material's potential for large-scale production upon application of this top-down approach.

## Results

**Morphological and chemical characterizations of wood films.** Cutting the natural wood along its longitudinal direction maintains the channel structure in the plane of the wood film (Fig. 2a). As shown in Fig. 2b, the microscopic wood channels with irregular polygonal shapes are well arranged along the longitudinal direction. A cross-sectional scanning electron microscopy (SEM) image shows that the thickness of the natural wood is 300 μm (Supplementary Fig. 2). After chemical delignification in a solution containing NaOH and $Na_2SO_3$ (see the Experimental Section for more details), the functional groups assigned to hemicellulose (-(C=O)-) and lignin (-(C=O)-O-) largely decrease compared to the starting material (Supplementary Fig. 3). The component analysis results also show that the lignin and hemicellulose are partially removed by the aqueous solution of $NaOH/Na_2SO_3$ (Supplementary Fig. 4). After partial removal of lignin and hemicellulose from the wood cell walls, the wood channels become soft and a significant amount of cellulose nanofibers are exposed in the cell wall surfaces (Supplementary Fig. 5).

Upon hot-pressing of this delignified material, a flexible and ultrathin wood film was obtained (Fig. 2c), with the porous wood completely converted to a densely packed laminated structure (Fig. 2d). The wood thickness was substantially reduced to 8.5 μm, indicating a reduction of 97% in thickness (Supplementary Fig. 6). Using SEM, we measured the thickness of the ultrathin wood film along its length at intervals of 5 μm and found it was uniformly below 10 μm, as shown in Fig. 2e. As a control, we measured the thickness of the natural wood without chemical treatment after hot-pressing and found it decreased from 300 μm to 100 μm under the same compression conditions, with many wood channels still present in the compressed wood structure (Supplementary Fig. 7). The sharp contrast in the microstructure and thickness between the compressed delignified wood and compressed natural wood highlights the critical role of the partial delignification treatment prior to densification.

Notably, the cellulose nanofibers in the ultrathin wood film remain highly oriented but are more densely laminated than in the natural wood starting material (Fig. 2f, g). X-ray diffraction (XRD) analysis also demonstrates the aligned cellulose molecular chains in the ultrathin wood film. The characteristic peaks around $2\theta = 16°$ and $22.6°$ are assigned to the diffraction peaks of the (110) and (200) planes of the cellulose crystals, respectively (Supplementary Fig. 8). Meanwhile, the small-angle XRD pattern shows the molecular alignment of the cellulose nanofibers (Fig. 2h). These combined results indicate that the partial delignification and densification treatments do not change the

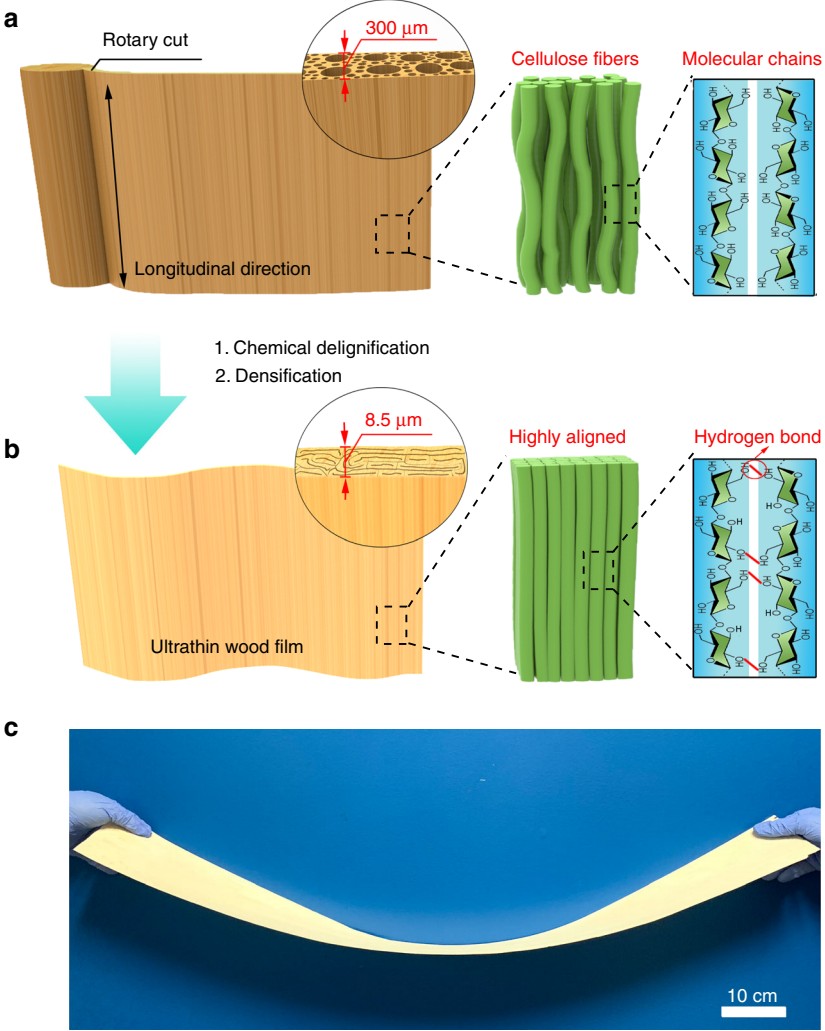

**Fig. 1** Schematic of the top-down approach to directly transform natural balsa into ultrathin wood films. **a** Left: schematic of the natural wood with its porous structure. Middle: the microstructure of the cellulose fibers in the wood cell walls. Right: the molecular chains of the cellulose fibers. **b** Left: schematic of the ultrathin wood with intertwined, compressed wood channels. Middle: the microstructure of the highly oriented cellulose fibers in the ultrathin wood cell walls. Right: hydrogen bond formation between the neighboring cellulose molecular chains. **c** Meter-long natural balsa wood with a thickness of 300 μm made in the lab

crystal structure or arrangement of cellulose nanofibers in the ultrathin wood film, which is important for the material's mechanical properties.

**Mechanical properties of wood films**. The resulting ultrathin wood film displays superb mechanical properties that stem from its highly oriented cellulose nanofibers and densely packed microstructure. We conducted mechanical tensile tests to evaluate the material's mechanical properties (Fig. 3a). Compared with natural wood, the ultrathin wood film shows greatly improved mechanical behavior, with a fracture strength and Young's modulus ($E$) of up to 342 MPa and 43.65 GPa, respectively (Fig. 3b, c). In contrast, the tensile strength and Young's modulus of the natural wood are only 17 MPa and 1.2 GPa, respectively. The ultrathin wood film shows almost 20-times improvement in tensile strength and 35-times enhancement in the Young's modulus. We observed the morphology of the fractured surfaces of the natural wood and ultrathin film using SEM to obtain further insights of the underlying mechanics. In the natural wood slice, the cross-section after the tensile test shows a porous microstructure with numerous wood channels (Fig. 3d). In this material, the cellulose fibers can be easily pulled out from the loosely assembled wood channels under tension, leading to a low fracture strength. In contrast, the wood cell walls in the ultrathin wood film are intertwined together after densification, which not only increases the interfacial area between the wood vessels but also benefits the formation of hydrogen bonds between aligned cellulose nanofibers (Fig. 3e). As a result, compared with natural wood, the firmly compressed cellulose nanofibers in the ultrathin wood film require more energy to be pulled out. The high tensile strength and Young's modulus of the ultrathin wood film far exceed that of typical plastic and natural biomaterials, further demonstrating its excellent mechanical properties (Fig. 3f). The ultrathin wood film also possesses great flexibility and foldability, which enables various origami designs (Fig. 3g–j). The ultrathin wood film can be folded into various shapes due to the material's ultrathin thickness and highly aligned cellulose nanofibers, while the natural wood is brittle and easily broken upon bending and folding (Fig. 3k–n). This folding behavior overcomes the limitations of traditional wood, which is generally unable to support the kind of processing required to form microdevices,

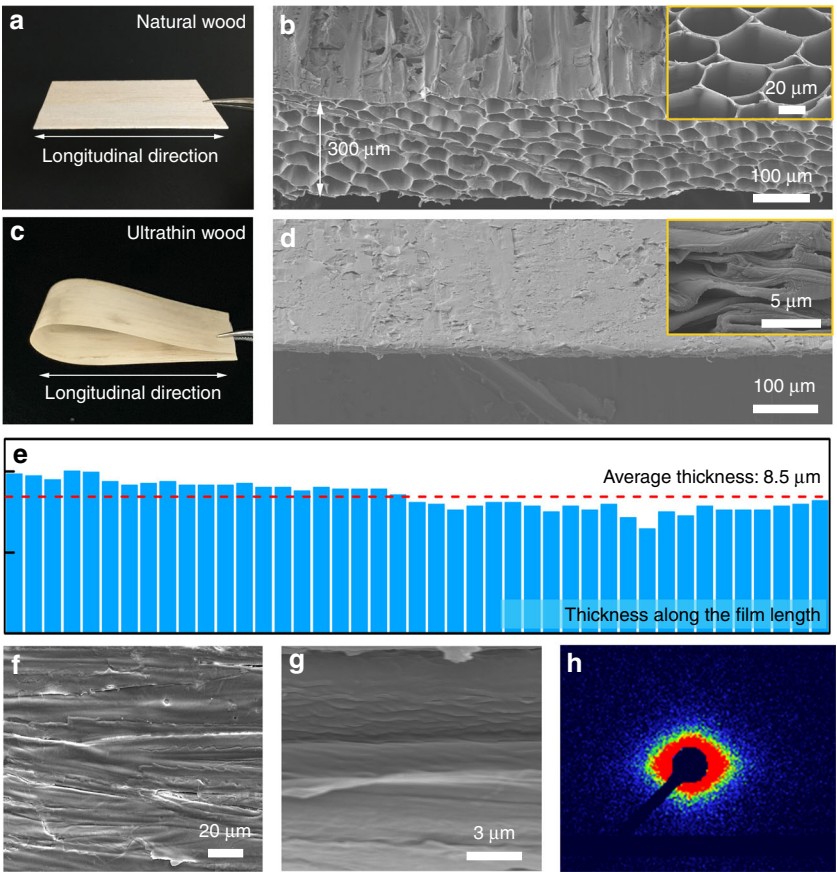

**Fig. 2** Morphological characterization of wood films. **a** Photograph of the rotary cut natural wood. **b** SEM image of the natural wood, with a thickness of 300 μm. Inset: top-view SEM image of the natural wood, showing its porous wood structure. **c** Photograph of the ultrathin wood. **d** SEM image of the ultrathin wood film, demonstrating its densified wood structure. Inset: Top-view SEM image of the ultrathin wood, revealing its collapsed wood cell walls. **e** The measured thickness of the ultrathin wood along its length at intervals of 5 μm, indicating uniform film thickness. **f**, **g** SEM images of the ultrathin wood, showing the aligned cellulose fibers. **h** Small-angle XRD pattern of the ultrathin wood, indicating the anisotropic alignment of the cellulose nanofibers

suggesting the great potential of using ultrathin wood films for photonics, acoustic sensors, and flexible electronics.

**Acoustic properties and vibration behaviors of wood films.** The ultrathin wood film can be an attractive alternative to conventional polymer films for high-performance acoustic transducers. In particular, the high Young's modulus and ultrathin thickness of the wood film should help to increase the resonance frequency and enhance the displacement amplitude of the diaphragm vibration, respectively[47]. We anticipate that the enhanced vibrational characteristics of our ultrathin wood film will make it highly suitable as a diaphragm for acoustic transducers with a wide operation bandwidth, high sensitivity (for microphones), and high sound pressure level (for speakers)[47–49].

To confirm the increase in the resonance frequency and displacement amplitude of the wood diaphragm, we characterized the frequency response of a vibrating circular diaphragm made of the ultrathin wood film that was 4.3 mm in diameter and compared it with a conventional polymer film (polypropylene) that was obtained from a commercial miniature speaker (Fig. 4a). As shown in Fig. 4b, the wood diaphragm with a thickness of 50 μm has a resonance frequency of 25.4 kHz, which is 1.83-times higher than that of the commercial polymer diaphragm with a thickness of 80 μm (13.9 kHz). It should be noted that the operation frequency bandwidth of diaphragm-based acoustic devices is often limited by its first resonance frequency. The

manufactured 50-μm-thick wood diaphragm with a first resonance frequency of 25.4 kHz can cover the entire range of audible frequencies for humans (20 Hz to 20 kHz). These resonances are the first resonance of the diaphragm vibration, which we confirmed by their vibration mode shapes (the (0, 1) vibration mode of a circular diaphragm), as shown in Fig. 4c, d. For the (0,1) vibration mode of a circular diaphragm with a fixed edge, the natural frequency of the diaphragm can be written as[47]

$$f_{01} = \frac{(\alpha_{201}a)^2}{2\pi a^2} \left[ \frac{Eh^2}{12\rho(1-\nu^2)} \right]^{1/2} \qquad (1)$$

in which $a$ and $h$ are the radius and thickness of the diaphragm, respectively, $\alpha_{201}a$ is a constant, and $E$, $\rho$, and $\nu$ are the Young's modulus, density, and Poisson's ratio of the diaphragm material, respectively. From Eq. (1), we can see that the first natural frequency of a circular diaphragm increases with the Young's modulus and thickness. Although the wood diaphragm is thinner than the polymer diaphragm, its much higher Young's modulus (more than 10-times higher) results in the observed increase in its natural frequency. On the other hand, the displacement amplitude of a circular diaphragm decreases with the increasing Young's modulus of the material. The decrease in the displacement amplitude can be compensated by reducing the diaphragm thickness. For example, given the Young's modulus and damping of a circular diaphragm, a thinner diaphragm renders a larger displacement amplitude, which can be explained by

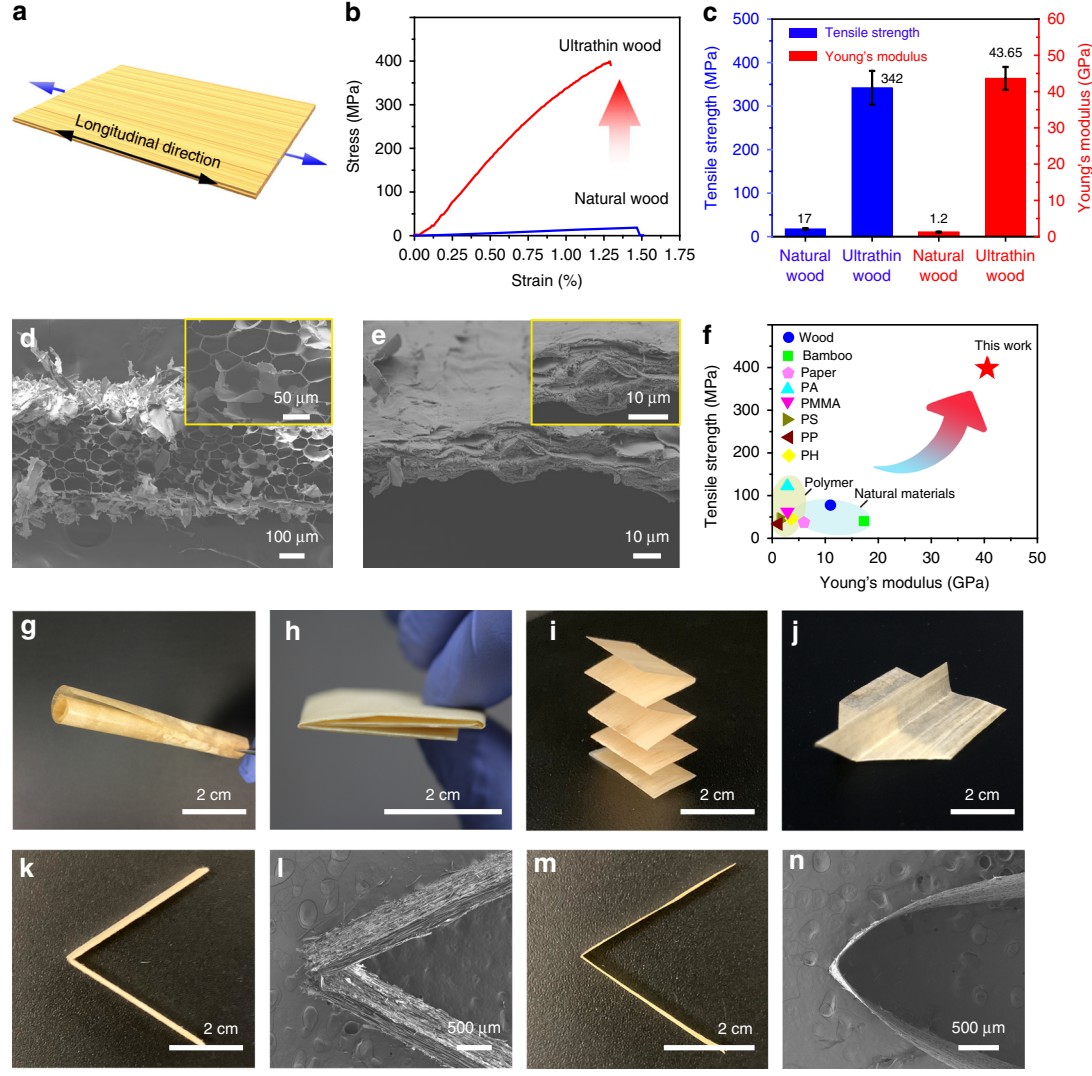

**Fig. 3** Mechanical properties of wood films. **a** Schematic of the tensile test along the longitudinal direction. **b** Corresponding tensile stress as a function of strain for the natural wood (blue line) and ultrathin wood film (red line). **c** Comparison of the tensile strength and Young's modulus of the natural wood and ultrathin wood film. Error bars represent standard deviation. **d, e** SEM images of the tensile fracture surface of the natural wood and ultrathin wood film. **f** Comparison of the tensile strength and Young's modulus of the ultrathin wood film with other widely used polymer and natural materials[50]. (PA: Polyamide; PMMA: Poly (methyl methacrylate); PS: Polystyrene; PP: Polypropylene) **g–j** Photographs of the ultrathin wood film demonstrating its flexibility and various origami designs. **k, l** Photograph and SEM image of the natural wood after bending, showing its rigid wood structure. **m, n** Photograph and SEM image of the ultrathin wood film after bending, showing its excellent flexibility and folding performance

Eqs. ((2)–(4)). Near the first natural frequency, the displacement amplitude at the diaphragm center ($r = 0$) can be expressed as[47]:

$$U_0 = \frac{2\pi pa}{\rho h N_1} \frac{\left[\frac{1}{\alpha_{201}} J_1(\alpha_{201}a) - \frac{J_0(\alpha_{201}a)}{\alpha_{101}I_0(\alpha_{101}a)} I_1(\alpha_{101}a)\right]\left[1 - \frac{J_0(\alpha_{201}a)}{I_0(\alpha_{101}a)}\right]}{\omega_1 \sqrt{\left[1 - \left(\frac{\omega}{\omega_1}\right)^2\right]^2 + 4\zeta_1^2\left(\frac{\omega}{\omega_1}\right)^2}}$$

(2)

$$N_1 = \int_0^a 2\pi r\left[J_0(\alpha_{201}r) - \frac{J_0(\alpha_{201}a)}{I_0(\alpha_{101}a)} I_0(\alpha_{101}r)\right]^2 dr$$

(3)

$$\zeta_1 = \frac{\mu}{\rho h \omega_1}$$

(4)

in which $p$ is the pressure amplitude, $\omega$ is the excitation frequency, $\omega_1$ is the first natural frequency, $\alpha_{101}a$ is a constant, and $\mu$ is the damping coefficient. The displacement amplitude of the 50-μm-thick wood diaphragm near the diaphragm center is 88.1 nm V$^{-1}$ at the resonance frequency, which is 1.25-times higher compared with the conventional polymer diaphragm (70.4 nm V$^{-1}$) (Fig. 4e). Furthermore, we can fabricate various thicknesses of wood films by adjusting the pressure during the hot-pressing process, which allows us to control the resonance frequency and displacement amplitude of the wood diaphragm. Figure 4f shows the frequency responses of 4.3 mm diameter circular wood diaphragms with different thicknesses. The natural wood with a thickness of 300 μm featured no resonance frequency in the range of 0 to 40 kHz (Supplementary Fig. 9). As the diaphragm thickness decreases from 80 μm to 30 μm, the first resonance frequency decreases from 26.2 kHz to 11.5 kHz, and the displacement amplitude at the resonance increases from 47.6 nm V$^{-1}$ to 188.9 nm V$^{-1}$ (Fig. 4f). Further reducing the thickness of the wood film diaphragm to 10 μm results in a displacement amplitude of 340 nm V$^{-1}$ and a first resonance frequency of 10.3 kHz (Supplementary Fig. 10). The corresponding displacement and first frequency of the wood films as function of

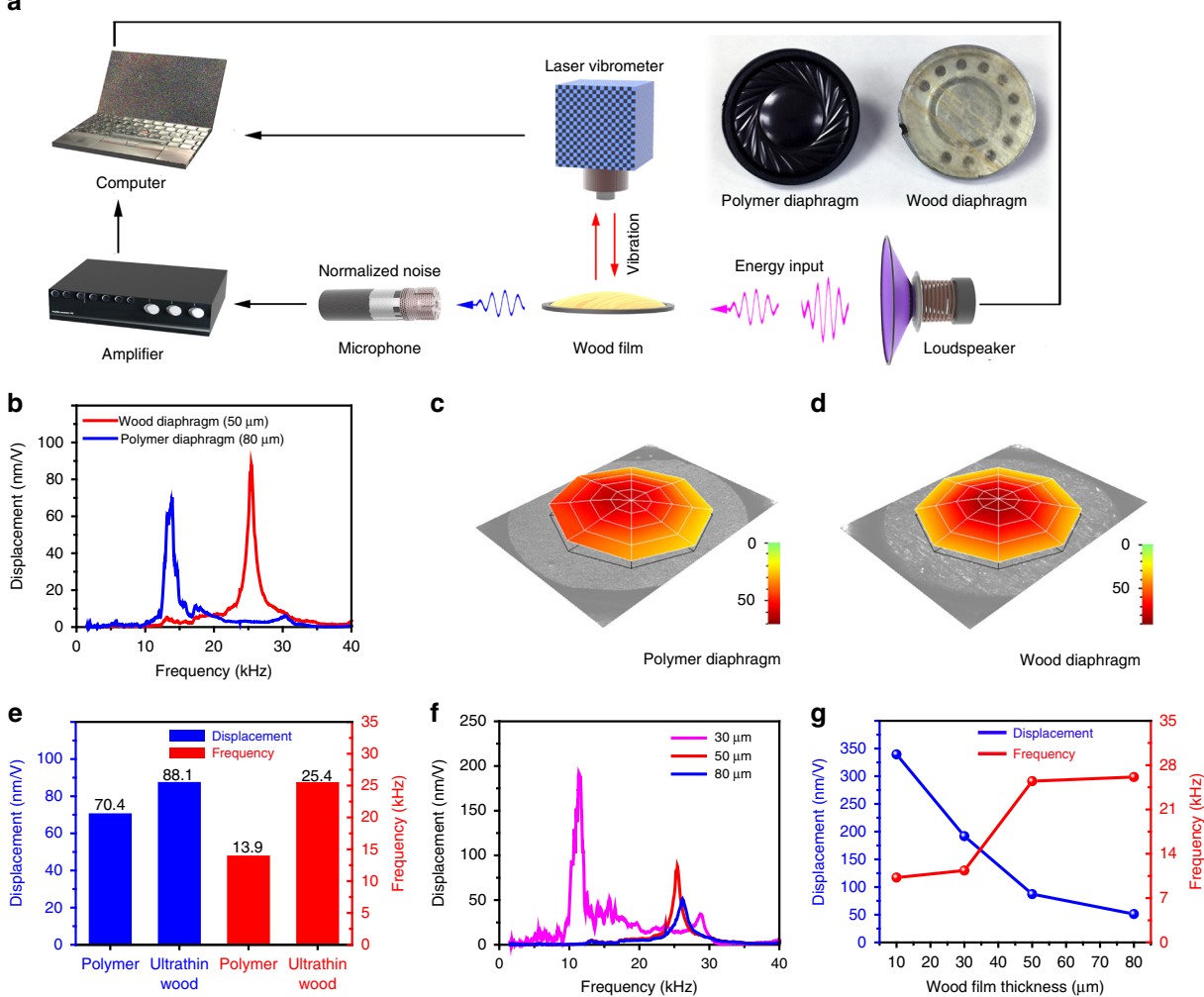

**Fig. 4** Acoustic properties of wood films. **a** Schematic of the vibrational frequency response measurement system. Insets: photographs of the commercial polymer diaphragm and ultrathin wood film. **b** The vibrational frequency response characteristics of the ultrathin wood film (50 μm) and the commercial polymer (80 μm) diaphragms. **c**, **d** The (0, 1) mode shapes of the polymer (80 μm) and ultrathin wood film (50 μm) diaphragms, respectively. **e** Comparison of the corresponding displacement and first resonance frequency of the ultrathin wood film (50 μm) and polymer (80 μm) diaphragms. **f** The vibrational frequency response characteristics of the ultrathin wood film at different thicknesses. **g** The corresponding displacement and first resonance frequency as a function of the thickness for the wood films

their thickness is shown in Fig. 4g, indicating that through proper selection of the wood film thickness, a wood diaphragm can provide higher resonance frequency as well as larger displacement amplitude than that of conventional polymer diaphragms, which is highly desirable for high-performance acoustic transducers.

For the tested circular wood diaphragm, although the directional cellulose fibers render the anisotropy in ultrathin wood film, the anisotropy does not affect the (0,1) mode vibration behaviors (Supplementary Fig. 11). The tested wood diaphragm under different direction has an almost identical frequency response regardless its longitudinal direction with respect to the sound wave excitation direction.

**Wood speaker prototype**. The diaphragm is a key component in commercial speakers, since their vibration performance determines the quality of the sound source. To demonstrate the promising application of the ultrathin wood film as an acoustic transducer, we assembled a miniature speaker made with the ultrathin wood diaphragm (Fig. 5a). Our prototype was composed of the miniature speaker made with the wood diaphragm (36 mm inner diameter) and a circuit board (Fig. 5b). The miniature speaker also contains a

copper coil (14 mm inner diameter) and permanent magnet. We bound the diaphragm and copper coil together and placed them in front of the permanent magnet (Supplementary Fig. 12). When the electric current flows through the copper coil, the direction of its magnetic field will rapidly change, while the permanent magnetic field remains constant. As a result, the electromagnetic forces act on the coil, causing the diaphragm to vibrate back and forth. Thus, the electrical signal is translated into an audible sound through the change in air pressure caused by the diaphragm. The audible sound of the speaker prototype was recorded by a microphone and the sound wave was analyzed through Adobe Audition CC. Compared with the original audio file, the speaker made by the ultrathin wood diaphragm can also play beautiful music (recording of a Spain Matador March, Supplementary Movie 1). Similarly, the sound wave recorded by the microphone from the wood diaphragm shows a similar waveform to that of the original sound wave, suggesting the intriguing application of the ultrathin wood film as a diaphragm in a loud speaker (Fig. 5c, d). Furthermore, this wood speaker may be further improved by the proper diaphragm structure design and the precise assemblies in existing industrial processing. Implementation of this speaker design using our ultrathin wood film may

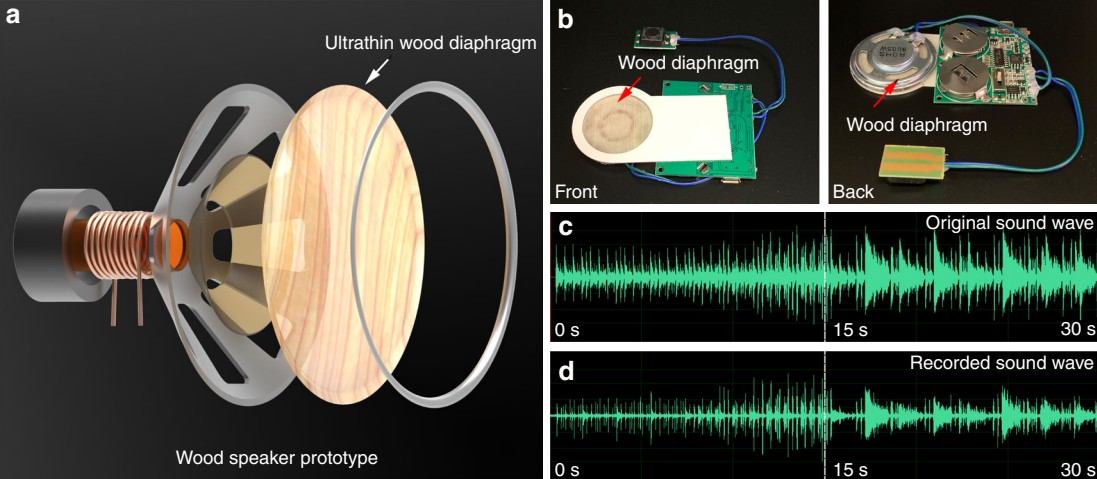

**Fig. 5** Wood speaker prototype. **a** Schematic of the wood speaker prototype. **b** Photographs of the speaker with the wood diaphragm. **c** The sound wave of the original song (Spain Matador March). **d** The recorded sound wave (Spain Matador March) of the speaker with the wood diaphragm

also compete with other technologies in the manufacture of microphones, hearing aids, and acoustic sensors.

## Discussion

We demonstrate an effective top-down strategy for the fabrication of an ultrathin wood film with a thickness of less than 10 μm directly from natural wood (a record-small thickness) via delignification and densification. The ultrathin wood film has a unique microstructure with intertwined wood cell walls and aligned cellulose nanofibers, which contribute to its outstanding mechanical properties with a more than 20-times increase in tensile strength (up to 342 MPa) and 35-times enhancement in Young's modulus (43.6 GPa). Compared with a commercial polypropylene diaphragm, using the strong and ultrathin wood film we can achieve a high-performance acoustic transducer with 1.83-times increased resonance frequency and 1.25-times enhanced displacement amplitude. We also demonstrate a loud speaker prototype that can generate music by the vibration of the ultrathin wood diaphragm, suggesting this material's great potential in acoustic applications, such as loud speakers, microphones, and hearing aids. The unique wood structure, ultrathin thickness, and remarkable mechanical properties can potentially enable many other micro designs as well, including wearable energy storage, ion-exchange membranes, food packaging, sensors, and catalyst supports. This demonstrated simple and scalable top-down approach may open up more potential functions and applications of strong film materials from abundant and biodegradable natural resources beyond plastic, metals, and ceramics.

## Methods

**Materials and chemicals**. Ultra-light balsa wood sheets with sizes of 10 cm × 10 cm × 0.4 mm were purchased from Specialized BALSA WOOD, LLC. The commercial speaker films were bought from Cylewet 2Pcs Loudspeaker via Amazon. The 120-grit sandpaper was purchased from LECO Corporation. Reagent grade NaOH (Sigma-Aldrich, reagent grade) and $Na_2SO_3$ (Sigma-Aldrich, reagent grade) were used for the partial removal of lignin from the wood.

**Fabrication process of the ultrathin wood film**. The natural balsa wood slices with sizes of 10 cm × 10 cm in width and length were polished to 0.3 mm thick using the 120-grit sandpaper. Then the wood slices were immersed and boiled in 2.5 M NaOH and 0.4 M $Na_2SO_3$ solution for 1 h. The processed wood slices were pressed at 100 °C under the pressure between 10 and 15 MPa for 24 h to obtain the ultrathin wood film.

**Mechanical testing**. The tensile properties and Young's Modulus of the wood samples were measured using a 30 kN Instron Testing Machine under ambient conditions. The dimensions for the natural balsa wood and ultrathin wood film samples were 100 mm × 5 mm × 0.4 mm and 100 mm × 5 mm × 0.01 mm,

respectively. The samples were clamped and stretched along the longitudinal direction and pulled apart at a constant speed of 5 mm min$^{-1}$.

**Vibration characteristics of the wood films**. Circular wood diaphragms were prepared by bonding the wood film to a M4 washer (inner diameter: 4.3 mm, outer diameter: 9.0 mm) using epoxy resin. The vibration of the wood diaphragms was characterized using scanning laser vibrometry. The prepared wood diaphragm was set up under the laser vibrometer (MSA-500, Polytec) and excited by white-noise sound with a 1.5–50 kHz frequency range, which was generated by a speaker (Petterson, LP 400). The frequency responses of the wood diaphragms were obtained by normalizing the vibrometer output against the output of a reference microphone (4191, Bruel & Kjaer) adjacent to the wood diaphragm. All tests were carried out under atmospheric conditions. The room temperature and the relative humidity is 24 °C and 50%, respectively.

**Miniature speaker prototype**. The circuit board and miniature speaker were purchased from Shenzhen Maiyout Technologu Co., Ltd (MY-H1650) (Supplementary Fig. 12). The product details are shown in Supplementary Fig. 13. The polypropylene diaphragm (36 mm in diameter) was replaced by the ultrathin wood film with a thickness of 50 μm (36 mm in diameter). Devcon epoxy resin (No. 14250) was used to bond the copper coils and wood diaphragm. The output sound was recorded by an iPhone XS placed at a distance of 10 cm from the miniature speaker. The sound wave was analyzed through Adobe Audition CC.

## Data availability

The data that support the findings of this study are available from the corresponding author upon reasonable request.

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

## Acknowledgements

We acknowledge the support of the Maryland NanoCenter and its AIM lab. We acknowledge partial funding for open access provided by the UMD Libraries' Open Access Publishing Fund.

## Author contributions

W.G., C.C. and H.K. contributed equally to this work. W.G. carried out the ultrathin wood preparing experiments. H.K. and M.Y. were responsible for both acoustic tests and calculations. Z.L., Z.D., Z.Z. and W.G. contributed to the miniature speaker tests. W.G., S.X., and C.C. measured the mechanical properties. J.D. and W.G. created the 3D illustrations. W.P. and S.H. provided characterization via SEM and FTIR. W.G., C.C., H.K., M.Y. and L.H. collectively wrote the paper. All authors commented on the final manuscript.

## Competing interests
The authors declare no competing interests.
