## [Peer Review File · Nature Communications]

Reviewers' comments:

Reviewer #1 (Remarks to the Author):

This is an interesting article. This reviewer only has a few minor comments listed below

1. Wood cell wall structure is directional as the authors noticed, while paper made of wood fibers can be made non-isotropic. Would this pose a problem for speaker application. Is there a prefer direction for this application.
2. Fig. 4, why the wood diaphragm did not show non-isotropic pattern, while the polymer diaphragm showed directional pattern?
3. Frequency response bandwidth is important for speakers. This information is missing in Fig. 4.

Reviewer #2 (Remarks to the Author):

This is a nice set of work building off the group's past efforts in using unique modified wood structures for novel electronic devices and uses.

- Their claims of developing unique wood based films of high mechanical performance is unique. The only related work that comes to mind, is older prior work on densification of wood, but is vastly different for much thicker wood panels and used for entirely different application area. Likewise, more recent work with paper or cellulose nanomaterials (CNM), are dealing with films produced by particles in suspension and require completely different processing route, and the film properties are much lower.
- The claims of this work is supported with the necessary data, and I have not seen anything like this in the prior literature.
- The authors work is novel, and of interest to the wider field, as it demonstrates the possibilities of producing high stiffness and high strength thin films from wood, which greatly expands the utility of wood outside of the traditional forestry and paper product industries. It opens up the possibilities of new materials utilization of renewable, sustainable, and biodegradable wood based materials.

Some comments/concerns with the paper that the authors should consider addressing or providing brief statements on.

1)A bit of semantics here, but I would recommend that the authors use the wood science terminology to describe wood structure. The authors use "wood growth direction", to define an axis within the anisotropic structure of wood, which I believe is aligned to the "tall" direction of the tree, as the wood channel structure they describe is in this orientation. Perhaps the readers can connect to this. However, from a wood science perspective this is grossly inaccurate, in which wood's anisotropic structure is defined by three axis: longitudinal (or axial), radial, and transverse (see attached schematic). The longitudinal direction is what the author means by "wood growth direction". The authors should use the wood science definition. Also, the tree does not only grow upward, seasonal tree growth rings is growth in the radial direction, resulting in trees increasing their diameter. This is not the direction that authors are referring to in this study, perhaps readers would get confused by this.

[See attached image of wood structure]

2)More description is needed on the film preparation. What was size of the wood slices used in hot pressing and what was the applied pressure?

3) Since cellulose based materials are typically moisture sensitive, some clarification on testing conditions are needed. Were samples held and then tested in ambient conditions?

4) The authors should mention something about property anisotropy that would result from the considerable structural anisotropy within their films. See figure 1b, the cellulose fibril structure is significantly more aligned in the longitudinal direction, versus the direction perpendicular (e.g., this is the tangential direction of the wood structure). The property anisotropy between longitudinal vs tangential direction within wood is well documented over that past 100+ years. A similar effect is well documented in paper physics with pulp fiber alignment effecting property anisotropy, and related studies using cellulose nanomaterials (CNM) that are highly aligned, have considerable property anisotropy when testing parallel vs transvers to the direction of CNM alignment. The issue for the current paper is how film property anisotropy would affect the flexural and acoustic response. For the tests used in this paper a circular speaker geometry is used, which may be more appropriate for testing films having property isotropy within the plan of the film. Interestingly, the authors could consider adjusting the speaker geometry (rectangular vs circular) and orientation of their films, to further optimize performance as compared to standard in plan isotropic materials. Just a thought....

Point-by-Point Response to Referees' Comments

(Black italic: Reviewer's remarks; Blue type: Our response)

Referee #1:

This is an interesting article. This reviewer only has a few minor comments listed below

Reply to the Referee: We thank Referee #1 for the positive comments on our design for single-digit-micrometer thickness wood speaker.

1. Wood cell wall structure is directional as the authors noticed, while paper made of wood fibers can be made non-isotropic. Would this pose a problem for speaker application. Is there a prefer direction for this application.

Reply to the Referee: We thank the referee for the valuable comments.

The sound quality of a speaker is the result of diaphragm materials and geometric designs.^[1-3] The ideal speaker diaphragm material needs to possess high Young's modulus, lightweight and good intrinsic damping. The natural wood film with anisotropic mechanical properties along the longitudinal direction have been used for speaker applications through appropriate geometric designs.^[1, 2] In this study, the pressed wood film with much improved mechanical properties (e.g. high Young's modulus) and ultrathin thickness (down to several micrometers, the thinnest wood product ever achieved) are anticipated to meet the materials requirement for high performance speaker and acoustic applications.

In our speaker applications with a circular wood diaphragm, the anisotropy of pressed wood film does not affect the behaviors of (0,1) mode vibration. When vibrating in (0, 1) mode, the membrane acts like a monopole source, which radiates sound effectively. As shown in Fig. R1, our wood diaphragm exhibits almost identical frequency responses regardless its longitudinal direction with respect to the sound wave excitation direction.

Figure R1. The (0, 1) mode shapes and vibrational frequency response characteristics of ultrathin wood film under sound wave excitations of different directions.

In the revised manuscript, Figure R1 is added to the supporting information as Supplementary Figure 11, and the above discussions are also added to the manuscript:

“For the tested circular wood diaphragm, although the directional cellulose fibers render the anisotropy in ultrathin wood film, the anisotropy does not affect the (0,1) mode vibration behaviors (Supplementary Fig. 11). The tested wood diaphragm under different direction has an almost identical frequency response regardless its longitudinal direction with respect to the sound wave excitation direction.” (Page 15)

2. Fig. 4, why the wood diaphragm did not show non-isotropic pattern, while the polymer diaphragm showed directional pattern?

Reply to the Referee: We appreciate the referee’s thoughtful comments.

In Figure 4c, the asymmetric (0, 1) mode shape of polymer is not a result of its anisotropic property. The slight shift of vibration center is caused by the fabrication variation. For example, asymmetric minor overflow of the epoxy resin of the diaphragm can slightly shift the vibration center from the supporting frame (i.e., M4 washer) center. Due to such a fabrication variation, the maximum amplitude of all films tested in this experiment is hard to appear at the center point. In Figure 4d, the maximum amplitude of wood film is also slightly shifted in the (0, 1) mode shape. As a result, we deleted the marked center point in Figure 4c and Figure 4d.

Revised Figure 4c and 4d. The (0, 1) mode shapes of the (c) polymer (80 μm) and (d) ultrathin wood film (50 μm) diaphragms, respectively.

3. *Frequency response bandwidth is important for speakers. This information is missing in Fig. 4.*

Reply to the Referee: We thank the referee for the valuable comments.

As shown in Figure 4b, the first frequency response peak of ultrathin wood film with the thickness of 50 μm is 25.4 kHz, which means the ultrathin wood film with the thickness of 50 μm used in speaker has the capability of output sound wave with the frequency from 20 Hz to at least 25.4 kHz. Such a large frequency bandwidth covers the entire range of audible frequencies for humans (20 Hz to 20 kHz). However, it should be noted that the bandwidth of a speaker does not only depend on the film material, but also on the overall speaker device design (e.g., shape of the membrane and resonance cavity), which will significantly change the resonance behavior of the speaker. Meanwhile, as shown in Figure 4g, the first resonant frequency of wood films decreases as the wood film thickness decreases, indicating the wood diaphragm with different thickness can provide different frequency response bandwidth in speaker. Proper selection of wood diaphragms with different thickness can help develop various speakers with different frequency response bandwidth. A wood diaphragm can provide higher resonance frequency (i.e., wide frequency bandwidth) as well as larger displacement amplitude than conventional polymer diaphragms, which is highly desirable for high-performance acoustic transducers.

The details are included in the revised manuscript:

“As shown in Fig. 4b, the wood diaphragm with a thickness of 50 μm has a resonance frequency of 25.4 kHz, which is 1.83 times higher than that of the commercial polymer diaphragm with a thickness of 80 μm (13.9 kHz). It should be noted that the operation frequency bandwidth of diaphragm-based acoustic devices is often limited by its first resonance frequency. The manufactured 50 μm thick wood diaphragm with a first resonance frequency of 25.4 kHz can cover the entire range of audible frequencies for humans (20 Hz to 20 kHz).” (Page 13)

Referee #2:

This is a nice set of work building off the group's past efforts in using unique modified wood structures for novel electronic devices and uses.

- *Their claims of developing unique wood based films of high mechanical performance is unique. The only related work that comes to mind, is older prior work on densification of wood, but is vastly different for much thicker wood panels and used for entirely different application area. Likewise, more recent work with paper or cellulose nanomaterials (CNM), are dealing with films produced by particles in suspension and require completely different processing route, and the film properties are much lower.*

- *The claims of this work is supported with the necessary data, and I have not seen anything like this in the prior literature.*

- *The authors work is novel, and of interest to the wider field, as it demonstrates the possibilities of producing high stiffness and high strength thin films from wood, which greatly expands the utility of wood outside of the traditional forestry and paper product industries. It opens up the possibilities of new materials utilization of renewable, sustainable, and biodegradable wood based materials.*

Reply to the Referee: We thank Referee #2 for their positive comments, especially on our novel single-digit-micrometer thickness wood design and its significance.

1) A bit of semantics here, but I would recommend that the authors use the wood science terminology to describe wood structure. The authors use "wood growth direction", to define an axis within the anisotropic structure of wood, which I believe is aligned to the "tall" direction of the tree, as the wood channel structure they describe is in this orientation. Perhaps the readers can connect to this. However, from a wood science perspective this is grossly inaccurate, in which wood's anisotropic structure is defined by three axis: longitudinal (or axial), radial, and transverse (see attached schematic). The longitudinal direction is what the author means by "wood growth direction". The authors should use the wood science definition. Also, the tree does not only grow upward, seasonal tree growth rings is growth in the radial direction, resulting in trees increasing their diameter. This is not the direction that authors are referring to in this study, perhaps readers would get confused by this. [See attached image of wood structure]

Reply to the Referee: We thank the referee for the careful comments.

Following the suggestion, we have revised the Figure 1, Figure 2 and Figure 3, and related descriptions by using the wood science terminology.

"Featuring aligned cellulose fibers embedded in a soft hydrogel matrix of lignin and hemicellulose within a porous channel structure that runs along the longitudinal direction (Fig. 1a)" (Page 4)

"Cutting the natural wood along its longitudinal direction maintains the channel structure in the plane of the wood film (Fig. 2a)." (Page 7)

"(a) Schematic of the tensile test along the longitudinal direction." (Page 11)

2) More description is needed on the film preparation. What was size of the wood slices used in hot pressing and what was the applied pressure?

Reply to the Referee: We thanks for the valuable comments.

We have added some details related to the preparation process of ultrathin wood film in revised manuscript. The size of wood slices used in hot pressing is 10 cm × 10 cm × 0.3 mm, and the applied pressure range is between 10 to 15 MPa.

“The natural balsa wood slices with sizes of 10 cm × 10 cm were polished to 0.3 mm thick using the 120-grit sandpaper. Then the wood slices were immersed and boiled in 2.5 M NaOH and 0.4 M Na₂SO₃ solution for 1 h. The processed wood slices were pressed at 100 °C under the pressure between 10 to 15 MPa for 24 h to obtain the ultrathin wood film.” (Page 18)

3) Since cellulose based materials are typically moisture sensitive, some clarification on testing conditions are need. Were samples held and then tested in ambient conditions?

Reply to the Referee: We thanks for the valuable comments.

Yes, the mechanical properties and acoustic tests of wood samples were measured under ambient conditions. The room temperature and the relative humidity is 24 °C and 50%, respectively.

“The tensile properties and Young’s Modulus of the wood samples were measured using a 30 kN Instron Testing Machine under ambient conditions.” (Page 18)

“All tests were carried out under atmospheric conditions. The room temperature and the relative humidity is 24 °C and 50%, respectively.” (Page 19)

4) The authors should mention something about property anisotropy that would result from the considerable structural anisotropy within their films. See figure 1b, the cellulose fibril structure is significantly more aligned in the longitudinal direction, verse the direction perpendicular (e.g., this is the tangential direction of the wood structure). The property anisotropy between longitudinal vs tangential direction within wood is well documented over that past 100+ years. A similar effect is well documented in paper physics with pulp fiber alignment effecting property anisotropy, and related studies using cellulose nanomaterials (CNM) that are highly aligned, have considerable property anisotropy when testing parallel vs transvers to the direction of CNM alignment. The issue for the current paper is how film property anisotropy would affect the flexural and acoustic response. For the tests used in this paper a circular speaker geometry is used, which may be more appropriate for testing films having property isotropy within the plan of the film. Interestingly, the authors could consider adjusting the speaker geometry (rectangular vs circular) and orientation of their films, to further optimize performance as compared to standard in plan isotropic materials. Just a thought....

Reply to the Referee: We appreciate the referee’s valuable comments.

In response to *“The authors should mention something about property anisotropy that would result from the considerable structural anisotropy within their films.”; “The issue for the current paper is how film property anisotropy would affect the flexural and acoustic response” and “For the tests used in this paper a circular speaker geometry is used, which may be more appropriate for testing films having property isotropy within the plan of the film”*

Regarding the structural anisotropy and its influence in flexural and acoustic response, Reviewer #1 has the similar comments. We have thoroughly addressed these comments on Pages 3-4 of this Response for Reviewer #1. In our speaker applications with a circular wood diaphragm, the anisotropy of pressed wood film does not affect the flexural and acoustic response (e.g., (0,1) mode vibration behaviors).

In response to *“have considerable property anisotropy when testing parallel vs transvers to the direction of CNM alignment” and “the authors could consider adjusting the speaker geometry (rectangular vs circular) and orientation of their films, to further optimize performance as compared to standard in plan isotropic materials”*

We agree with the referee that the pressed wood films have considerable property anisotropy when testing parallel vs. transverse to the direction of CNM alignment. To evaluate whether such mechanical property anisotropy will affect the acoustic performance of the pressed wood film, we carried out additional vibration characteristics experiments by changing the angles between the longitudinal direction of wood and the sound wave propagation direction. As shown Figure R1, all circular wood diaphragm shows an almost same frequency response regardless its longitudinal direction with respect to the sound wave propagation direction, demonstrating the directional wood samples does not affect the vibrational frequency response behaviors.

Due to the limitation of testing setup, we can only evaluate circular samples. In-depth analysis about the effects of rectangular and circular wood diaphragm to acoustic response behaviors will be carried out in our future studies.

References

- 1 Kageyama, T. and Suzuki, K., Yamaha Corp, 1994. Acoustic diaphragm. U.S. Patent 5,329,072.
- 2 Imamura, S., Hirano, T., Ogata, T. and Kuwahata, T., Victor Co of Japan Ltd, 2008. Speaker diaphragms, manufacturing methods of the same, and dynamic speakers. U.S. Patent 7,467,686.
- 3 Kim, Y.N., KH Chemical Co Ltd, 2008. Acoustic Diaphragm and Speaker Having the Same. U.S. Patent Application 12/090,348.

REVIEWERS' COMMENTS:

Reviewer #1 (Remarks to the Author):

The authors satisfactorily addressed my comments

Reviewer #2 (Remarks to the Author):

The authors addressed my comments adequately.